# Study on Low Temperature Cracking Resistance of Carbon Fiber Geogrid Reinforced Asphalt Pavement Surface Combined Body

**DOI:** 10.3390/polym16152168

**Published:** 2024-07-30

**Authors:** Zhiqiang Wang, Guangqing Yang, Xin Wang, Xunmei Liang, Mengfan Liu, Hao Zhang

**Affiliations:** 1State Key Laboratory of Mechanical Behavior and System Safety of Traffic Engineering Structures, Shijiazhuang Tiedao University, Shijiazhuang 050043, China; 15195850272@163.com; 2School of Traffic and Transportation, Shijiazhuang Tiedao University, Shijiazhuang 050043, China; 3School of Civil Engineering, Shijiazhuang Tiedao University, Shijiazhuang 050043, China; 1202201194@student.stdu.edu.cn (M.L.); 1202201013@student.stdu.edu.cn (H.Z.); 4School of Urban Geology and Engineering, Hebei GEO University, Shijiazhuang 050031, China; wangx@stdu.edu.cn; 5Shandong Road New Materials Co., Ltd., Tai’an 271000, China; ludelxm@163.com

**Keywords:** road engineering, surface combined body, carbon fiber geogrid, low-temperature bending damage test, low-temperature cracking resistance

## Abstract

Currently, there are limitations in the research on the use of carbon fiber geogrids to prevent low-temperature cracking in asphalt pavements. This study aims to comparatively investigate the effects of carbon fiber-based geogrid type and dense-graded asphalt concrete mixture (AC) surface combined body (SCB) type on the low-temperature cracking resistance of reinforced asphalt pavement through low-temperature bending damage tests. Two geogrid types were prepared: a carbon fiber geogrid (CCF) and a glass/carbon fiber composite qualified geogrid (GCF). Two SCB types were studied: AC-13/AC-20 and AC-20/AC-25. The results show that the improvement in the flexural tensile strength of CCF is similar to that for GCF. Moreover, under reinforced conditions, the improvement in the low-temperature cracking resistance of AC-20/AC-25 is better than that for AC-13/AC-20 by 16.26–24.57%. Based on the analysis, the reasonable ratio range of the aperture sizes to the major particle sizes in the dense gradation can achieve a more effective interlocking effect. This can improve the low-temperature cracking resistance of carbon fiber-based geogrid-reinforced samples. Then, increasing the bending absorption energy is a key way of improving the low-temperature cracking resistance of carbon fiber-based geogrid reinforcements. Eventually, the fracture type of carbon fiber-based geogrid-reinforced samples is a mixed plastic–brittle fracture. These results can provide a reference for the road failure analysis of geogrid-reinforced asphalt pavement.

## 1. Introduction

Cracking is one of the main distresses of asphalt pavement, accelerating the damage to asphalt pavement and seriously shortening its service life [1,2,3]. Low-temperature cracking is one of the major causes of cracking in asphalt pavements [4,5,6]. In response to the cracking problem of asphalt pavement, many scholars have conducted a series of research works on the low-temperature cracking resistance of asphalt mixtures [7,8,9] but have failed to solve the cracking problem fundamentally and effectively. Nowadays, geosynthetic-reinforced structures with excellent technical performance and obvious economic advantages have been widely used in railways, highways, and other transport infrastructures [10,11,12,13].

Ali Khodaii et al. [14] investigated the influence of temperature, geogrid position, and existing surface layer type on the anti-reflective crack performance of a geogrid-reinforced asphalt overlay by conducting type I fracture tests. The results showed that the rate of crack propagation in the reinforced specimens was significantly reduced with respect to that in the unreinforced specimens. Wang et al. [15] evaluated the cracking resistance of interlayer materials by temperature-type reflection cracking (flexural-tensile type) test methods and found that the anti-cracking effect of a reinforced fiberglass geogrid was better than that of an unreinforced geogrid. Fereidoon Moghadas Nejad et al. [16] analyzed the effect of geosynthetics and their moduli on increasing the loading times required for crack initiation and propagation by conducting cyclic loading tests on different types of geosynthetic-reinforced asphalt mixtures. The results showed that the rate of crack propagation was related to the modulus of the geosynthetics. Irene Gonzalez-Torre et al. [17,18] studied the effect of the geosynthetic type and secant modulus on the crack opening by applying high- and low-frequency loads to account for traffic and thermal effects and showed that the presence of geosynthetics effectively hindered the propagation of cracks. Fereidoon Moghadas Nejad et al. [19] investigated the influence of temperature, type of geosynthetics, and crack width on the improvement of the anti-reflective cracking performance of an asphalt overlay based on the response surface methodology. The results revealed that geogrids were effective in improving the performance of overlay layers against reflective cracking systems. Sireesh Saride et al. [20] studied the performance of geosynthetic-reinforced asphalt overlays against crack propagation under a four-point bending fatigue test by means of the digital image correlation (DIC) technique and showed that the inclusion of geosynthetics hindered the propagation of reflective cracks but led to delamination problems. Jong-Hoon Lee et al. [21] investigated the long-term performance of fiber geogrid-reinforced asphalt overlay pavements by means of finite element analysis (FEA), laboratory testing, and field investigation, which showed that laying carbon fiber geogrids at the bottom of the overlays can effectively delay the crack propagation at the interface. Jianming Ling et al. [22] researched the ability of three types of interlayer structures to resist reflective cracking, including atactic polypropylene modified linoleum, geotextiles, and stress-absorbing layers with the newly reflective cracking test method and found that geotextiles had the optimum performance. Ehsan Solatiyan et al. [23] investigated the mechanical properties of reinforced asphalt overlays by the three-point bending test on cylindrical specimens. In addition, some studies have shown that geogrid reinforcing materials did not have a substantial effect on the process of crack initiation in asphalt mixtures but only on the process of crack propagation, delaying crack propagation [24,25,26].

Asphalt composition, fiber quality, and aging are also factors that affect the low-temperature cracking of asphalt mixtures and pavements [27,28,29]. Ren et al. [30] studied the effect of aging degree on the relaxation behavior of asphalt and proposed that long-term aging increases the relaxation time, shear stress, and residual stress ratio of asphalt. Lin et al. [31] investigated the effects of lignin fibers and polyester fibers on the cracking resistance of high-content SBS-polymer-modified asphalt (HCPMA) mixtures before and after aging and found that both fibers exhibited more significant strengthening effects after short-term and long-term aging.

From the above literature review, it can be noted that most of the studies focus on the cracking resistance of geosynthetic-reinforced asphalt pavement, especially on the cracking resistance of glass fiber geogrid-reinforced asphalt pavements. However, the cracking resistance of carbon fiber-based geogrid-reinforced asphalt pavements has rarely been studied. Due to the high production cost of carbon fiber, its widespread application is limited to certain high-end fields [32]. One of the main reasons for the high cost of carbon fibers is the high cost of the precursor yarn [32,33], which accounts for more than 50% of the total production cost of carbon fibers. One method to reduce the cost of carbon fiber is to use low-cost precursors of textile-grade PAN fibers (Tex PAN) [32,34]. In recent years, some researchers have successfully prepared Tex PAN-based carbon fibers through fiber treatment before and after stabilization [32,34,35]. In particular, the influence of the performance difference of longitudinal and transverse ribs on the cracking resistance of carbon fiber-based geogrid-reinforced asphalt pavement has rarely been examined. The carbon fiber-based geogrid is a new type of warp-knitted geogrid with excellent mechanical properties, high- and low-temperature resistance, good compatibility with asphalt mixtures, and good engineering characteristics, and it is not easily affected by temperature.

The surface structure of asphalt pavement is generally divided into the upper layer, the intermediate surface layer, and the lower layer. According to the two-body model [36,37,38], the surface combined body (SCB) model is the upper layer–intermediate layer model and the intermediate layer–lower layer model, respectively. Such surface combined bodies are more in line with the actual asphalt pavement structures and facilitate construction.

Based on the above analysis, the aim of this paper is to understand in depth the influence of the difference in longitudinal and transverse rib performance of carbon fiber-based geogrids and changes in the surface structure on the low-temperature cracking resistance of asphalt pavements. In addition, this paper proposes that the fracture energy of asphalt pavements at low temperatures is composed of bending absorption energy and toughness energy. Therefore, this paper focuses primarily on understanding the influence of carbon fiber-based geogrid type (referred to as geogrid type) and dense-graded asphalt concrete mixture (AC) surface combined body type (referred to as surface combined body type) on the low-temperature cracking resistance of reinforced asphalt pavement. Based on the low-temperature bending damage test, first, the effects of geogrid type and SCB type on a single low-temperature cracking resistance parameter are analyzed in depth. Second, the influences of geogrid type and SCB type on the comprehensive low-temperature cracking resistance parameters are discussed. Finally, the crack propagation patterns in carbon fiber-based geogrid-reinforced asphalt pavement surface combined body are revealed by an interface macro analysis. The research in this paper contributes to an improved understanding of the low-temperature cracking resistance of reinforced asphalt surface structures.

## 2. Materials and Methods

### 2.1. Materials

The geogrids used as reinforcements for this experimental study are a carbon fiber geogrid (CCF) and a glass/carbon fiber composite qualified geogrid (GCF). The transverse ribs of the GCF are glass fiber transverse ribs, and the longitudinal ribs of the GCF are carbon fiber longitudinal ribs, as shown in Figure 1. The geogrids were provided by Shandong Road New Materials Co., Ltd (Tai’an, China). which were used as received. The technical specifications of the geogrids were obtained through tests in accordance with the Chinese standard of JTG E50 [39], as shown in Table 1. The longitudinal ultimate tensile strength of the glass/carbon fiber composite grids is low compared with that of the carbon fiber geogrids.

The asphalt used in this test is SBS (styrene–butadiene–styrene block copolymer)-modified asphalt. Properties of the SBS-modified asphalt were obtained through tests in accordance with the Chinese standard of JTG E20 [40], as shown in Table 2. The standardized requirements are derived from the Chinese standard of JTG F40 [41]. The asphalt binder used in the AC mixtures is SBS-modified emulsified asphalt, whose dosage is 0.4 L/m^2^. The top layer is an AC-13 asphalt mixture consisting of basalt and limestone aggregate with a nominal maximum size of aggregate of 13 mm. According to some actual conditions of asphalt pavement in China [42,43,44], the aggregates with nominal particle sizes of 10–15 mm and 5–10 mm are basalt, while the rest of the aggregates are limestone. The middle surface layer is an AC-20 asphalt mixture with limestone aggregate with a nominal maximum size of aggregate of 20 mm. The bottom layer is an AC-25 asphalt mixture made up of limestone aggregate with a nominal maximum size of aggregate of 25 mm. Properties of the basalt and limestone were obtained through tests in accordance with the Chinese standard of JTG E42 [45], as shown in Table 3. Limestone mineral powder is utilized as filler material. The aggregate gradation curves for each mixture type are shown in Figure 2. The asphalt aggregate ratios of the three asphalt mixtures (AC-13, AC-20, and AC-25) were determined by the Marshall test, and the test results are shown in Table 4.

### 2.2. Specimen Preparation

In order to compare and analyze the influence of geogrid type and SCB type on the low-temperature cracking resistance of asphalt pavement surface combined bodies, a series of low-temperature bending damage tests were conducted. The reinforced surface combined bodies were chosen as the target structure to be analyzed, and the unreinforced SCB (UN) was selected as the control group. Two geogrid types were selected, namely the glass/carbon fiber composite qualified geogrid (GCF) and the carbon fiber geogrid (CCF). Two SCB types were chosen, which were the upper layer–intermediate layer (AC-13/AC-20) and the intermediate layer–lower layer (AC-20/AC-25). The above combined bodies were designed for different interface positions of the actual asphalt surface structure, simulating the rutting performance of asphalt mixtures at different interface locations.

The SCB specimens of reinforced asphalt pavement used in this study were prismatic-shaped, with dimensions of 250 × 47 × 50 mm. Steel molds with the bulk dimensions of 300 × 300 × 100 mm and 300 × 300 × 50 mm were utilized to prepare the specimens. First, in accordance with the requirements of the Chinese standard of JTG E-20 [40] (Code of China 2011), the lower asphalt mixture was mixed and rolled into shape. Second, the lower molded specimen was put into a large test mold, and the surface was cleaned and uniformly coated with the SBS-modified emulsified asphalt, and the geogrid was laid. Then, the upper asphalt mixture was rolled and molded using a similar method to that for the lower asphalt mixture. Finally, the rutting slabs were cut into sections to prepare all the prismatic specimens. The specimen preparation procedure is shown in Figure 3. The layout form of the reinforced surface combined body (RSCB) is shown in Figure 4. For reinforced specimens, the tests were conducted in the direction of the longitudinal ribs, and the performances of the longitudinal ribs were the same. The main difference between the two lies in the direction of the transverse ribs, which have different strengths and stiffness.

### 2.3. Low-Temperature Bending Damage Test

For evaluating low-temperature cracking resistance, a series of low-temperature bending damage tests were conducted using the WD-402 high- and low-temperature test chamber and the WDW-1020 micro-controlled electronic universal testing machine as illustrated in Figure 5. The specimens were subjected to the displacement loading mode with a loading rate of 50 mm·min^−1^ and a temperature of −10 °C. The temperature was derived from the Chinese standard of JTG E20 [40]. The test termination conditions were defined by crack propagation to 80% of the height of the specimens. Six groups of tests were carried out. Five replicates were set up for each group test. The mean value was quoted as the result of the test. The result met the permissible coefficient of variation of 20%. The effects of the geogrid type and SCB type on the low-temperature cracking resistance of asphalt pavement combined bodies were investigated by means of the relevant parameters of the test.

### 2.4. Interface Observations

Macroscopic analysis was used for interface observations. By taking photos to visually observe and analyze the bending failure specimens, the crack propagation path and fracture type of the specimens were determined.

## 3. Results

### 3.1. Effect of Reinforcements on Flexural Tensile Strength

Figure 6 shows the variation curves of flexural tensile stress with loading time for the UN and the RSCB. As can be seen from Figure 6, the flexural tensile stress–loading time curves of the SCBs can be roughly divided into three stages: the first stage is a slow-growth stage, which shows that the flexural tensile stress increases slowly with the increase in loading time, and the test duration is relatively short. The second stage is the seemingly linear growth stage with the longest test duration. It shows that the flexural tensile stress increases rapidly with the increase in loading time until the peak flexural tensile stress is reached. The third stage is the post-peak drop stage with a short test duration. It shows that the peak flexural tensile stress drops linearly and rapidly with the increase in loading time, and even the flexural tensile stress is almost zero.

It can also be seen from Figure 6 that the flexural tensile stress of the carbon fiber-based geogrid reinforcement both increase. The time to reach the peak value is gradually prolonged. The flexural tensile strength is also gradually increased, which is greater than that of the UN. In the case of AC-13/AC-20 (AC-20/AC-25) combination, the time to peak for the GCF and the CCF is prolonged by 54.5–70.8% and 65.5–88.0%, respectively, compared to the UN.

According to the lamination theory [46], the potential reinforcing effect of laying carbon fiber-based geogrids between the layers of the SCBs is applied to increase the modulus of the upper layer [47], which increases the stiffness ratio of the upper and lower layers of the SCBs. The larger the stiffness ratio of the upper and lower layers of the SCB, the stronger the stress diffusion effect. The slower the rate of stress growth at the bottom of the beam, the longer the stress time required to reach cracking at the bottom of the beam. The greater the load required to reach ultimate stress at the bottom of the beam, the greater the flexural tensile strength. The presence of carbon fiber-based geogrids also reduces the compression–tension stiffness ratio of the SCB. The stress diffusion effect increases as the compressive–tensile stiffness ratio of the SCB decreases. It can be seen that the ultimate tensile strength and stiffness of carbon fiber-based geogrids play an important role in improving the flexural tensile strength of the RSCBs.

As the overall ultimate tensile strength of the carbon fiber-based geogrid increases, the drop rate of the peak flexural tensile stress of the reinforcement decreases, and the residual flexural tensile stress gradually increases. The degree of the post-peak damage brittleness is strong, and the flexural tensile stress drops obviously, showing similar post-peak damage characteristics. That is because the geogrid has a restraining effect on the crack opening through its own high tensile strength, interlayer bonding, and frictional resistance. So that the RSCB has a certain residual flexural stress.

From the above analysis, it can be seen that the flexural tensile strengths of different types of geogrid RSCBs are higher than those of UNs. In order to better compare the flexural tensile strength changes in different types of geogrids after reinforcement under different SCB types to reflect the effect of reinforcement, the strength enhancement factor (*SEF*) is introduced:(1)$$SEF=\frac{R_{B}^{*}-R_{B}}{R_{B}}\times 100,$$
where *R_B_*^*^ is the flexural tensile strength of the RSCB (in Mpa), and *R_B_* is the flexural tensile strength of the UN (in Mpa).

The flexural tensile strength results of the SCBs are presented in Figure 7. The *SEF*s of different geogrid RSCBs can be calculated by Equation (1), as shown in Table 5.

The following can be seen from Table 5 and Figure 7:

The *SEF*s of different geogrid RSCBs are greater than 0 under different SCB types. The *SEF* of the CCF RSCB is as high as 25.24% in the case of AC-20/AC-25, which shows the positive effect of the carbon fiber-based geogrid in improving the strength of the SCB. The positive effect of the CCF is higher than that of the GCF.

Under the same SCB conditions, with the increase in the overall ultimate tensile strength of the carbon fiber-based geogrid, the *SEF* increases. In the case of the AC-20/AC-25 (AC-13/AC-20) combination, the *SEF* increased by 5.07–5.16% when the geogrid was changed from GCF to CCF, indicating that the performance differences in the longitudinal and transverse ribs of carbon fiber-based geogrids play a certain role in increasing the *SEF* of the RSCB.

For the same reinforcement conditions, the flexural tensile strengths of the AC-20/AC-25 combination are higher than those of the AC-13/AC-20 combination. Under UN conditions, the flexural tensile strength increases by 5.92% when the SCB changes from AC-13/AC-20 to AC-20/AC-25. Under the condition of GCF(CCF) reinforcement, the flexural tensile strength increases by 11.54–11.88% when the SCB changes from AC-13/AC-20 to AC-20/AC-25. It can be seen that the SCB type has an important contribution in improving the flexural tensile strength of the RSCB.

The reason lies in the fact that the coarse aggregate of the AC-20/AC-25 combination is mainly concentrated in 9.5–19 mm aggregate particles, referred to as the major particle size, which is close to the aperture size of the carbon fiber geogrid. The ratio range of the aperture sizes to the major particle sizes in the AC-20/AC-25 is 1.32–2.63, consistent with previous research findings [48,49,50,51,52]. When the aggregate particles are embedded in the openings, the ribs of the geogrid can restrict the free movement of the aggregate particles to better perform the restraining effect and dissipate the temperature stress generated by the low-temperature shrinkage. Meanwhile, the coarse aggregate of the AC-13/AC-20 combination is mainly composed of 4.75–13.2 mm aggregate particles. The ratio range of the aperture sizes to the major particle sizes in the AC-13/AC-20 combination is 1.89–5.26. The aggregate particles are small enough to go through the apertures, even as multiple particles together. The degree of the embedded locking of aggregate particles is improved to be lower, and the restraining effect of the geogrid is lower. Therefore, the ratio range of the aperture sizes to the major particle sizes in the dense gradation is an important factor in improving the flexural tensile strength of carbon fiber-based geogrid RSCBs.

### 3.2. Effect of Reinforcements on Flexural Tensile Strain

The variation curves of flexural tensile strain with loading time for the UN and the RSCB are shown in Figure 8a,b. Figure 8c,d show the variation curves of flexural tensile strain growth ratio with loading time for different SCBs. The flexural tensile strain growth ratio refers to the ratio of flexural tensile strain at a certain moment to the maximum flexural tensile strain at the time of failure. The growth rate refers to the slope of the curve showing the growth ratio of the flexural tensile strain over time. As can be seen from Figure 8, the flexural tensile strain and the growth ratio of the flexural tensile strain of the SCBs increase approximately linearly with the increase in loading time. The growth rates and the growth ratios of the flexural tensile strain of the RSCBs are lower than those of the Uns, suggesting that reinforcement of the carbon fiber geogrid can retard the growth of the flexural tensile strain of the SCB.

It can also be seen from Figure 8 that the growth rate and growth ratio of flexural tensile strain of CCF reinforcement are lower than those of GCF reinforcement. That is because the RSCB can be approximated as elastomers at low temperatures, and the flexural tensile strain is directly proportional to the flexural tensile strength. The growth rate of flexural tensile strength of the RSCB decreases from the GCF to the CCF about 100 ms later.

The maximum flexural tensile strain results of the SCBs are presented in Figure 9. As can be seen from Figure 9, the effect of geogrid type on the maximum flexural tensile strain of the SCBs is obvious. The maximum flexural tensile strain of the CCF reinforcement is the highest and that of the UN is the lowest at low temperatures, regardless of the SCB type. Compared with the UN, the maximum flexural tensile strain increases by 72.70–93.11% for CCF reinforcement and 50.38–51.36% for GCF reinforcement. This indicates that the laying of a carbon fiber-based geogrid between the layers improves the low-temperature flexibility of the SCB. The increasing effect of the CCF is better than that of the GCF. Due to the performance difference between the longitudinal and transverse ribs of the GCF, uneven stress distribution occurs at the interlayer interface under loading. Compared with CCF, the GCF has a weaker restraining effect on the aggregates and the reinforcing effect is reduced, producing smaller flexural tensile strains.

It can also be seen from Figure 9 that the SCB type has an effect in increasing the maximum flexural tensile strain of the RSCBs. The increasing effect of AC-20/AC-25 is better than that of AC-13/AC-20 at low temperatures, regardless of the geogrid type. The increasing effect of the maximum flexural tensile strain of the CCF reinforcement improves by 12.65% from AC-13/AC-20 to AC-20/AC-25. The reason may lie in the fact that under low-temperature conditions, the asphalt pavement reinforced surface layer combination can be approximated to an elastomer. As the increment in flexural tensile strength of the RSCB increases, so does the increment in flexural strain.

### 3.3. Relationship between Flexural Tensile Stress and Flexural Tensile Strain

To further analyze the relationship between the flexural tensile strain and flexural tensile stress, changes in the flexural tensile stress under different reinforcement conditions are plotted in Figure 10. As can be seen from Figure 10, the relationship curves between flexural tensile stress and flexural tensile strain of SCB specimens are similar to the Type I stress–strain curves in Figure T 0715-1 of the Chinese standard of JTG E-20 [40] (Code of China 2011), as shown in Figure 11a. The trends of the curves are similar, showing a basically linear increase in flexural tensile stress versus flexural tensile strain. After reaching the peak value, it decreases sharply, exhibiting obvious strain softening. The flexural tensile stress shows a basically linear decay relationship with the flexural tensile strain. From the viewpoint of the slope of the curve, the slope of the curve of the UN is the largest, followed by that of the GCF, and the smallest is for the CCF at low temperatures. It can be seen that the carbon fiber-base geogrid reinforcement can make the flexural stiffness of the SCB decrease obviously, and the linear correlation coefficient is large. The reason is that the laying of geogrids between layers allows the specimen to be subjected to a greater load for a longer period of time, which produces a greater deformation of the specimen and results in a lower flexural stiffness. The sharp type of the peak of the curve has not undergone a significant transformation. There is no yielding zone. The behavior is brittle in the UN and also in the RSCB.

### 3.4. Effect of Reinforcements on Fracture Energy

When evaluating the low-temperature cracking resistance of asphalt mixtures in the literature [53,54,55], the fracture energy (*G_f_*) is considered a comprehensive performance index that takes into account the strength and deformation of the material. Therefore, the fracture energy was used to evaluate the low-temperature cracking resistance of carbon fiber-based geogrid RSCBs. In order to better analyze the reinforcing effect of carbon fiber-based geogrids, the fracture energy (*G_f_*) was divided into two parts: the bending absorption energy (*Q*_B_) and the toughness energy (*G*_t_) [56]. The *Q*_B_ refers to the energy required for cracks to appear in the asphalt pavement SCB under the action of external forces. The *G*_t_ refers to the ability of the asphalt pavement SCB to continue to open and expand under external forces after cracks appear, until a degree of instability is reached, causing fractures.

For the modified load–displacement curves, the load is used to integrate the vertical deformation. The integration range of the *Q_B_* is from *d*_1_ to *d*_2_ [57,58]. The integration range of the *G*_t_ is from *d*_2_ to *d*_3_. The integration curves are shown in Figure 11b. Figure 12 shows the load–deflection curves of different SCBs. The *Q_B_* and *G_t_* values of the SCBs are calculated by Equation (2) and Equation (3), respectively, and the results are shown in Table 6.
(2)$$Q_{B}=\frac{1}{bh_{1}}\int_{d_{1}}^{d_{2}}P_{B}\left(x\right)d\left(x\right),$$
(3)$$G_{t}=\frac{1}{bh_{2}}\int_{d2}^{d_{3}}P_{B}\left(x\right)d\left(x\right),$$

In which *Q_B_* is the bending absorption energy of the specimen, J/m^2^; *h*_1_ is the height of the specimen, m; *b* is the width of the specimen, m; *d*_1_ is the modified initial deflection, m; *d*_2_ is the deflection corresponding to the maximum load after correction, m; *G_t_* is the toughness energy of the specimen, J/m^2^; *h*_2_ is the length of the crack propagation, m; *d*_3_ is the deflection corresponding to the termination of the test after correction., m; and *P_B_* is the maximum value of the load, N.

The following can be seen from Table 6:

The geogrid type has a significant effect on the low-temperature cracking resistance of the asphalt pavement SCB. Under low-temperature conditions, the *G_f_* of the CCF is the largest, followed by that of the GCF, and that of the UN is the smallest, regardless of the SCB type. The differences in the *G_f_* of the three reinforcements are significant under the same SCB conditions. Compared with the *G_f_* of the UN, the *G_f_* of the GCF and the CCF increases by 66.75–78.22% and 100.39–129.46%, respectively. It can be seen that carbon fiber geogrid reinforcement improves the low-temperature cracking resistance of the SCB. Compared with it, there is still a certain gap of the glass/carbon fiber composite qualified geogrid reinforcement.

For the same SCB type, the variability of the *G_f_* of the RSCB depends mainly on the variability of the *Q_B_* of the RSCB. Under the condition of the same SCB, the increment in the *Q_B_* of CCF reinforcement accounts for about 87.94–90.10% of the increment in the *G_f_* of CCF reinforcement, while the increment in the *G*_t_ of CCF reinforcement only accounts for about 9.90–12.06%. At the same time, the increment in the *Q_B_* of GCF reinforcement accounts for about 86.00–94.17% of the increment in the *G_f_* of GCF reinforcement, while the increment in the *G*_t_ of GCF reinforcement only accounts for about 5.82–14.00%. It is evident that increasing the *Q_B_* is a key element in improving the low-temperature cracking resistance of RSCBs.

For the AC-13/AC-20 (AC/20/AC-25) combination, the *Q_B_* of the asphalt pavement SCB under different geogrid reinforcement conditions is not significantly different and the situation of the *G*_t_ is also similar. However, the *Q_B_* and *G*_t_ increase with the increase in the overall ultimate tensile strength of the carbon fiber-based geogrids. When the reinforcement material is changed from GCF to CCF, the *Q_B_* increases by 18.18–30.27% and the *G*_t_ increases by 19.18–40.54%. It can be seen that the effect of the CCF is superior to that of the GCF from the energy point of view. That is because the planar grid-like structure of carbon fiber-based geogrids is formed by the intersection of longitudinal ribs and transverse ribs. The transverse ribs play an important role in maintaining the planar grid-like structure and improving the flexural rigidity of the longitudinal ribs [59]. The reduction of transverse rib performance changes the geometry of the GCF mesh, leading to a reduction in stiffness, mesh stiffness, and tensile friction resistance, which changes the reinforcement mechanism of the GCF in the SCB and reduces the reinforcing effect. Therefore, the performance difference between the longitudinal and transverse ribs plays an important role in enhancing the low-temperature cracking resistance of carbon fiber-based geogrid reinforced asphalt pavement SCBs.

The SCB type has a certain effect on the low-temperature cracking resistance of asphalt pavement SCB. Under reinforced conditions, the improvement in low-temperature cracking resistance of AC-20/AC-25 is better than that of AC-13/AC-20, by 16.26%-24.57%. The reason lies in the fact that the improvement in the RSCB is influenced by factors such as aggregate type, aggregate particle size, and the stiffness ratio of the upper and lower layers. Due to the presence of the geogrid, the more the aggregate particle size matches the grid size, the stronger the interlocking effect between the geogrid and the particles. In this situation, the higher the stiffness ratio of the upper and lower layers, the stronger the stress diffusion effect, and the stronger the improvement in low-temperature cracking resistance.

In addition, as can be seen from Figure 10, the toughness is enhanced, the brittleness decreases accordingly, and the crack propagation is delayed. The flexural tensile displacement significantly increases. However, it still tends to the brittle material fracture characteristics at low temperatures.

### 3.5. Interface Observations

The bending failure phenomena of the SCB specimens are depicted in Figure 13 and Figure 14.

As can be seen from Figure 13 and Figure 14, the crack propagation path of the UN specimen rises tortuously, forming a transverse crack of an overall “N-type”, in accordance with the results obtained by Zhang Xiaojing et al. [60]. On the other hand, the crack propagation path of the RSCB generally shows as the “non-N-type”, forming non-penetrating cracks. Meanwhile, the crack openings of the UNs are relatively large. However, the crack openings of the CCF and GCF RSCBs are relatively small, and the RSCBs do not fracture.

The reason lies in the fact that crack propagation will be blocked and redirected due to the presence of geogrids. The process of crack propagation and the change in direction is accompanied by stress redistribution and energy release, which reduces the degree of singularity at the crack tip and retards the upward expansion of the crack. In low-temperature conditions, the flexural tensile strain of the asphalt mixture is small. When the opening of the crack tip does not reach the deformation limit of the grid but reaches the cracking limit of the upper layer, the crack will continue to expand upward. In the process of expansion, the opening of the cracks is mainly restrained by the “bridging action” of the longitudinal ribs of the geogrid. The specimen eventually forms a non-penetrating crack with a small degree of opening. In addition, when the crack passes through the geogrid, the concentrated force is generated at the intersection of the geogrid and the crack, which reduces the stress intensity factor at the tip of the crack and slows down the expansion of the crack. Consequently, carbon fiber-base geogrids play a very important role in changing the crack propagation mode of asphalt pavement SCBs.

Regardless of the SCB type, the difference in the crack propagation path between CCF and GCF reinforcement is small. The difference between the CCF and GCF fails to significantly change the crack propagation pattern of the RSCB. Therefore, the transverse ribs of carbon fiber-based geogrids play a major role in changing the crack propagation pattern of the RSCB.

In the SCB, there are differences in the crack propagation paths between the upper and lower layers. The variability is large for AC-13/AC-20 and small for AC-20/AC-25, which is probably due to the performance of the aggregates. The aggregates used in AC-13 are basalt and limestone aggregates, in which the coarse aggregates are basalt aggregates. The aggregates used in both AC-20 and AC-25 are limestone aggregates. During the crack propagation process, the cracks easily pass through the limestone. Basalt is a highly wear-resistant material mineral with better crush, abrasion, and wear values than limestone. During crack propagation, it is difficult for the cracks to pass through the basalt; rather, they pass along the edge of the basalt, changing and extending the crack propagation path. Hence, aggregate properties are also an important factor in changing the crack propagation pattern of the RSCB.

The phenomena of “fracture along the stone” and “fracture through the stone” have occurred in carbon fiber-based geogrid-reinforced specimens. It has been shown that the fracture type of the carbon fiber-based geogrid RSCBs under a tension–compression load is a mixed plastic–brittle fracture where plastic fracture and brittle fracture coexist [61], which has certain significance for the road failure analysis of geogrid-reinforced asphalt pavement.

## 4. Discussion

Under low-temperature conditions, the RSCB increases the range of the SCB to withstand the load, changes the stress transfer path, and increases the vertical strain. The RSCB also improves the bending damage absorption energy, increases the toughness, and effectively slows down the crack extension, thus improving the low-temperature cracking resistance of the SCB. The results of this study are consistent with the research results of Fereidoon Moghadas Nejad [19], Adam Zofka [24], and Francesco Canestrari [26], among others. In addition, the RSCB prolonged the time for crack initiation and changed the crack propagation pattern compared to the UN. The research results provide an effective method for practically solving the low-temperature cracking problem of asphalt pavements.

For future studies, it is suggested that more geogrid types and SCB types be taken into consideration to validate the findings of this study. Furthermore, the interlayer bonding mechanism of the carbon fiber-based geogrid-reinforced asphalt pavement SCBs should be taken into account.

## 5. Conclusions

In this study, through a series of low-temperature bending damage tests, the influence of geogrid type and SCB type on the low-temperature cracking resistance of asphalt pavement SCB was investigated in depth. Based on the interface macro analysis, the crack propagation patterns in carbon fiber-based geogrid-reinforced asphalt pavement SCBs were revealed. The following conclusions can be made accordingly:(1)The flexural tensile strength and maximum flexural strain of carbon fiber-based geogrid reinforcement both increase, while the flexural stiffness decreases accordingly. The performance difference in the longitudinal and transverse ribs plays a certain role in improving the low-temperature cracking resistance of carbon fiber-based geogrid reinforcement. The improvement in flexural tensile strength of the CCF is similar to that of the GCF. Geogrid reinforcement extends the time to reach cracking by 54.5–88.0%.(2)The SCB type has an important influence in improving the flexural tensile strength and maximum flexural tensile strain of carbon fiber-based geogrid reinforcement. In this test, under reinforced conditions, the improvement in the low-temperature cracking resistance of AC-20/AC-25 is better than that of AC-13/AC-20, by 16.26–24.57%.(3)The ratio range of the aperture sizes to the major particle sizes in the dense gradation is an important factor in improving the low-temperature cracking resistance of carbon fiber-based geogrid reinforcements. In this test, when the ratio range is 1.32–2.63, the improvement in the low-temperature cracking resistance of carbon fiber-based geogrid reinforcement is more significant.(4)The relationship curve of flexural tensile stress versus flexural tensile strain for carbon fiber-based geogrid reinforcement is similar to the Type I stress–strain curve. During the test, there is an obvious strain softening phenomenon, but no obvious yield zone.(5)The fracture energy of carbon fiber-based geogrid reinforcement increases with the increase in the overall ultimate tensile strength of geogrid. Fracture energy consists of bending absorption energy and toughness energy, with bending absorption energy accounting for more than 80%. Therefore, improving the bending absorption energy is an important way to improve the low-temperature cracking resistance of carbon fiber-based geogrid reinforcements.(6)The crack propagation path of unreinforced geogrids generally shows as an “N-type”, while the crack propagation path of carbon fiber-base geogrid reinforcements is generally a “non-N-type”. The transverse ribs play a major role in changing the crack propagation pattern of carbon fiber-based geogrid reinforcements. The fracture type of carbon fiber-based geogrid reinforcements is a mixed plastic–brittle fracture, which can provide a reference for the road failure analysis of geogrid-reinforced asphalt pavement.

## Figures and Tables

**Figure 1 polymers-16-02168-f001:**
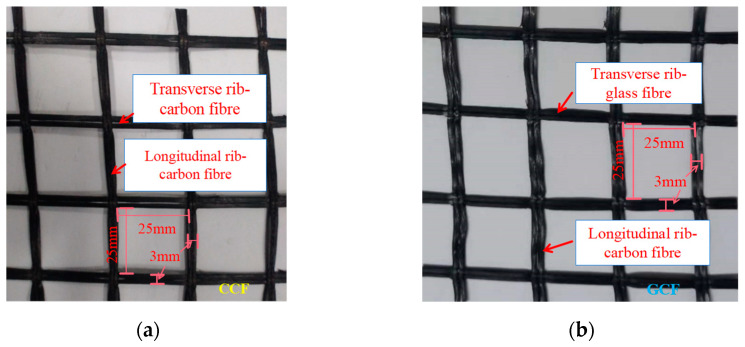
Types of geogrids: (**a**) carbon fiber geogrids; (**b**) glass/carbon fiber composite qualified geogrid.

**Figure 2 polymers-16-02168-f002:**
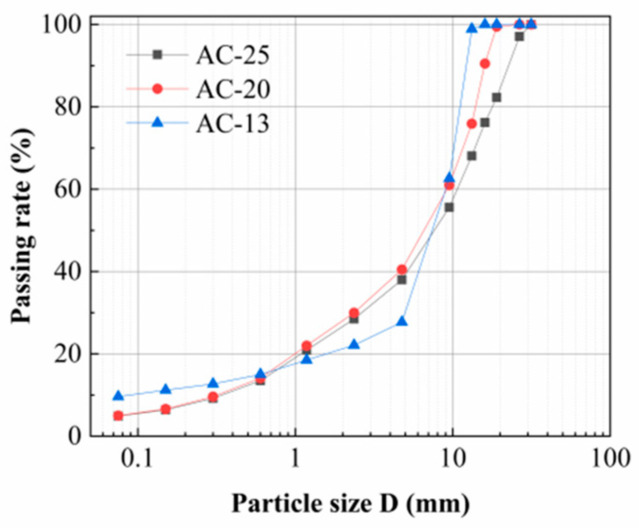
Asphalt mixture aggregate gradation curve diagram.

**Figure 3 polymers-16-02168-f003:**
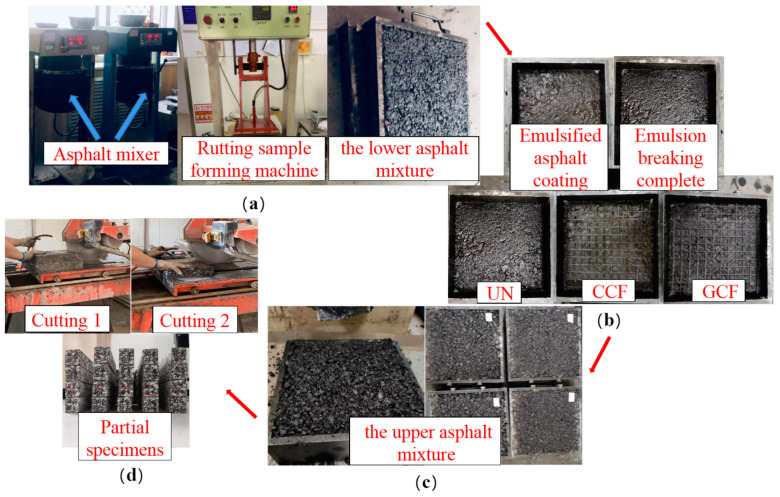
Specimen preparation procedure: (**a**) mixing and rolling to form the lower asphalt mixture; (**b**) laying the geogrid; (**c**) mixing and rolling to form the upper asphalt mixture; (**d**) cutting and preparing the specimens.

**Figure 4 polymers-16-02168-f004:**
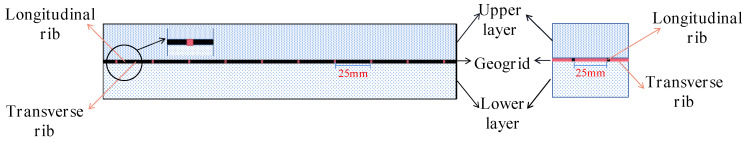
Layout form of the RSCB.

**Figure 5 polymers-16-02168-f005:**
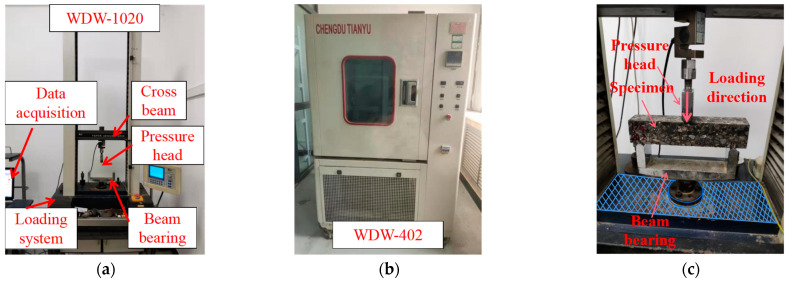
Test equipment: (**a**) micro-controlled electronic universal testing machine; (**b**) high- and low-temperature test chamber; (**c**) low-temperature bending damage test.

**Figure 6 polymers-16-02168-f006:**
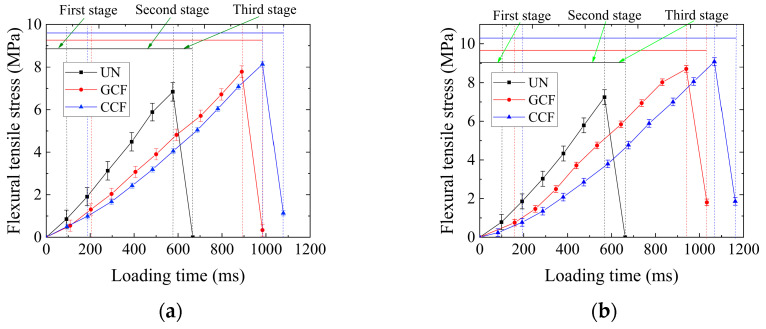
The variation curve of flexural tensile stress of the SCB with loading time: (**a**) AC-13/AC-20; (**b**) AC-20/AC-25.

**Figure 7 polymers-16-02168-f007:**
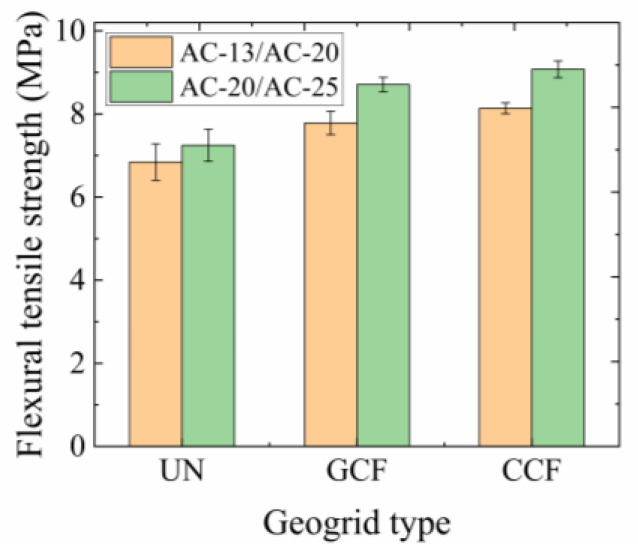
Flexural tensile strength of the SCBs.

**Figure 8 polymers-16-02168-f008:**
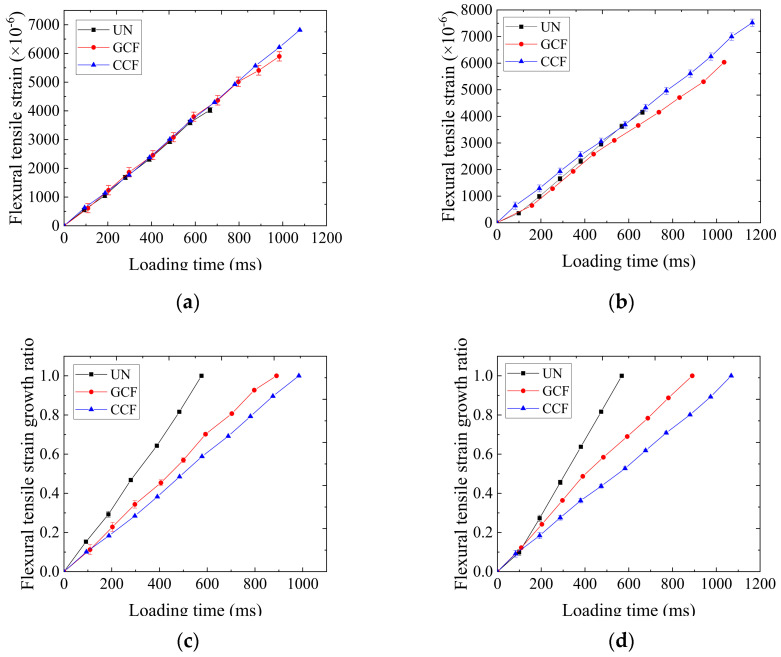
Curves of flexural tensile strain and flexural tensile strain growth ratio of SCB with loading time: (**a**) AC-13/AC-20; (**b**) AC-20/AC-25; (**c**) AC-13/AC-20; (**d**) AC-20/AC-25.

**Figure 9 polymers-16-02168-f009:**
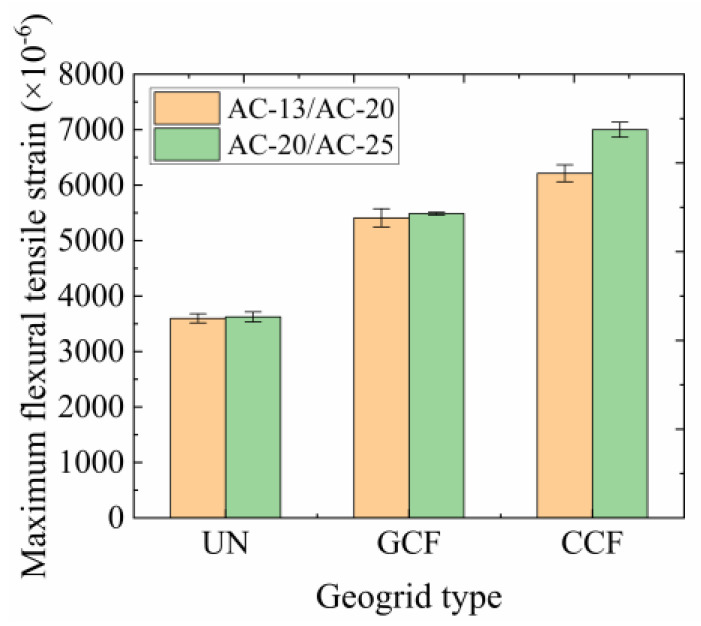
Maximum flexural tensile strain of the SCB.

**Figure 10 polymers-16-02168-f010:**
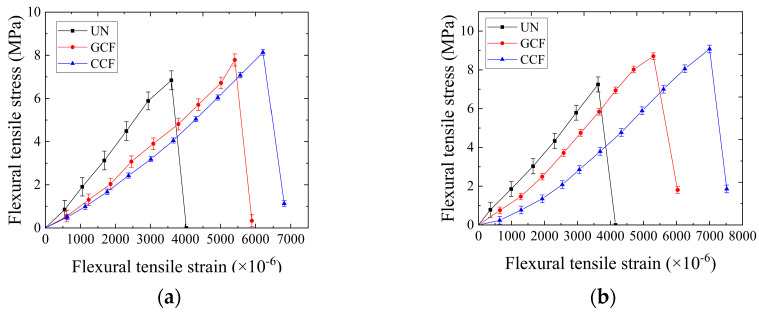
Curves showing the flexural tensile stress and flexural tensile strain of SCB specimens: (**a**) AC-13/AC-20; (**b**) AC-20/AC-25.

**Figure 11 polymers-16-02168-f011:**
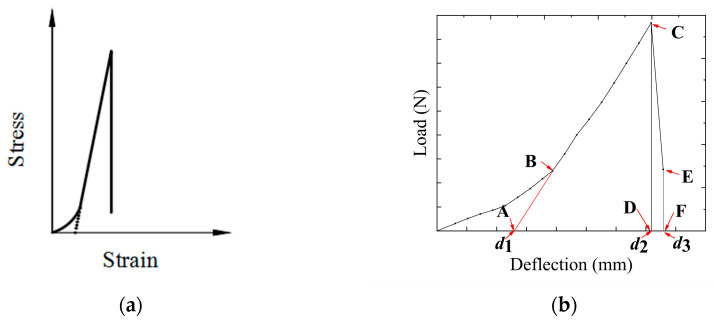
(**a**) Type I stress–strain curves [40]; (**b**) Load-deflection curve of the low-temperature bending damage test.

**Figure 12 polymers-16-02168-f012:**
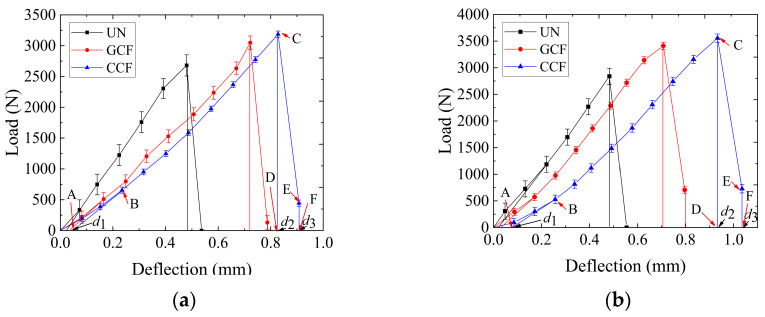
Load–deflection curve of SCBs: (**a**) AC-13/AC-20; (**b**) AC-20/AC-25.

**Figure 13 polymers-16-02168-f013:**
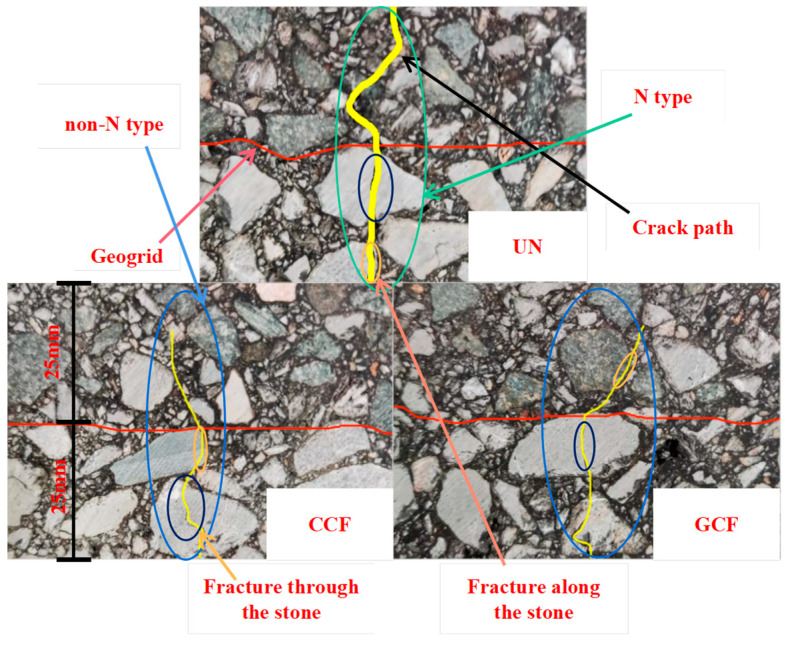
The bending failure phenomena of the AC-13/AC-20 combination.

**Figure 14 polymers-16-02168-f014:**
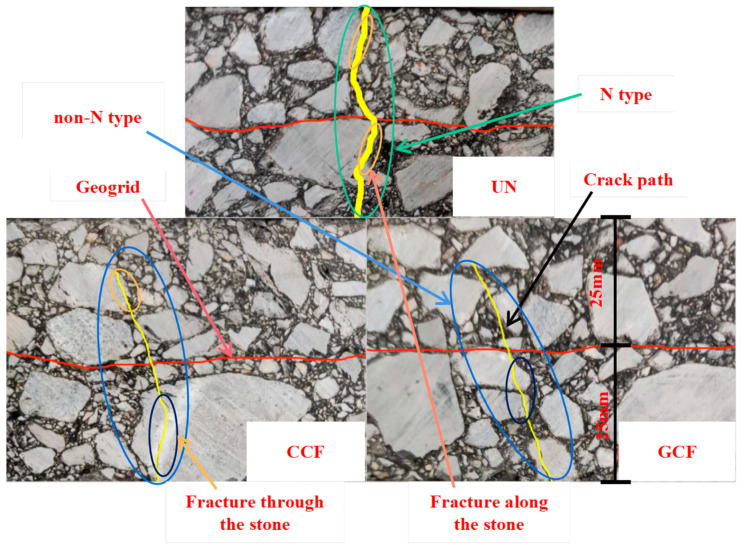
The bending failure phenomena of the AC-20/AC-25 combination.

**Table 1 polymers-16-02168-t001:** Technical parameters of the geogrids.

Index		GCF	CCF
Ultimate tensile strength (kN/m)	Transverse	50	80
	Longitudinal	80	80
Ultimate elongation(%)	Transverse	≤3	≤2
	Longitudinal	≤2	≤2
Thickness (mm)		0.6	0.6
Aperture size (mm × mm)		25 × 25	25 × 25

**Table 2 polymers-16-02168-t002:** Major physical characteristics and technical requirements of the SBS-modified asphalt.

Items	Measured Value	Standardized Requirements
Penetration, 25 °C, 5 s, 100 g (0.1 mm)	57.8	30~60
Ductility, 5 cm/min,10 °C (cm)	25.1	≥20
Softening point (°C)	64.4	≥60

**Table 3 polymers-16-02168-t003:** Major characteristics and technical requirements of the basalt and limestone.

Items	Measured Value	Standardized Requirements
	Basalt/Limestone	
Crushed stone value (%)	8.67/18.77	≤26
Polished stone value (PSV)	41.4/39.5	≥38
Los Angeles wear value (%)	7.60/26.89	≤28

**Table 4 polymers-16-02168-t004:** Asphalt–aggregate ratio of the asphalt mixtures.

Asphalt Mixture	AC-13	AC-20	AC-25
Asphalt–aggregate ratio (%)	4.8	4.3	3.7

**Table 5 polymers-16-02168-t005:** *SEF*s of the RSCB at failure(%).

SCB Type	GCF	CCF
	*SEF_CCF_* (%)	*SEF_GCF_* (%)
AC-13/AC-20	13.77	18.93
AC-20/AC-25	20.17	25.24

**Table 6 polymers-16-02168-t006:** Comprehensive indicators of the SCBs (unit: J/m^2^).

SCB Type	Geogrid Type								
	UN			GCF			CCF		
	*G_f_*	*Q_B_*	*G_t_*	*G_f_*	*Q_B_*	*G_t_*	*G_f_*	*Q_B_*	*G_t_*
AC-13/AC-20	294.78	262.28	32.50	491.56	447.58	43.98	590.71	528.90	61.81
AC-20/AC-25	320.68	277.80	42.88	571.50	493.51	77.99	735.83	642.88	92.95

## Data Availability

The data used to support the findings of this study are available from the corresponding author upon request.

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
