# Peer review of "Study on Low Temperature Cracking Resistance of Carbon Fiber Geogrid Reinforced Asphalt Pavement Surface Combined Body"

_polymers, 2024, doi:10.3390/polym16152168_

Round 1

Reviewer 1 Report

Comments and Suggestions for Authors

Dear authors,

Good topic of research. I found ~60 major and minor corrections that need to be made to make the article suitable for the journal and easy to read for its readers. Plese find my comments in the PDF attached. I have highlighted and commented on what and how can the manuscript can be improved. For now, it would be a major revision.

Comments on the Quality of English Language

Need some improvements. Look at my comments. I have suggested a few changes for improving quality of English as well. 

Author Response

Thank you for your comments concerning our manuscript entitled “Study on Low Temperature Cracking Resistance of Carbon Fiber Geogrid Reinforced Asphalt Pavement Surface Combined Body”.Please see the attachment.

Reviewer 2 Report

Comments and Suggestions for Authors

Polymers

Study on low temperature cracking resistance of carbon fiber geogrid reinforced asphalt pavement surface combined body

Comments:

This study presents an interesting work regarding the low-temperature cracking performance of carbon fiber geogrid reinforced asphalt pavement surface. The topic and results are interesting and useful for the long-life asphalt pavement design and construction. Here are some suggestions for further improve the quality of this manuscript:

1. How about the cost of using carbon fiber compared to natural fiber?

2. Introduction: The effects of bitumen composition, fiber quality, and aging on the low-temperature cracking of asphalt mixture and pavement can also be discussed here, such as: * Towards critical low-temperature relaxation indicators for effective rejuvenation efficiency evaluation of rejuvenator-aged bitumen blends. Journal of Cleaner Production. * Unraveling the influence of fibers on aging susceptibility and performance of high content polymer modified asphalt mixtures. Case Studies in Construction Materials.

3. Table 2: Please mention which standard used for the standardised requirement here.

4. It is not necessary to show the experiment plan in “Material and methods”.

5. Please combine the sections of “3. Results” and “4. Discussion”.

6. Can the authors add the error bar in figures, such as Figure 8.

7. Did the authors do some microstructure test and analysis on fracture surface?

8. Some recommendations for future work can be added in “Conclusion” section.

Comments on the Quality of English Language

Minor check and improvement is required.

Author Response

(The authors gave the same response as above.)

Round 2

Reviewer 1 Report

Comments and Suggestions for Authors

Please find the attachment of PDF with my comments. Some of the comments raised in the first review were missed. Those comments are raised again. 

Also, a big flaw in data processing is detected, after correcting this flaw lot of result and conclusion will change significantly. So the authors are requested to revise and re-read the whole manuscript (top to bottom) before submitting for the next round of review. 

Comments on the Quality of English Language

Lot of English error are highlighted but not commented anything in the PDF attachment above.

Several big sentences are noted in the manuscript. Reviewer suggest making smaller and easy-to-read sentences instead of such big half paragraph sentences. Some of them are pointed exclusively. Others may have not been marked, but should be changed to make it easy to read. 

Author Response

(The authors gave the same response as above.)

Round 3

Reviewer 1 Report

Comments and Suggestions for Authors

Great improvements are noted in the quality of the manuscript over last reviews. Now, I only see a few minor issues here and there as marked in the attached PDF. Please correct them. 

Comments on the Quality of English Language

English polishing is needed. 

I still see a lot of long sentences which make it very difficult to read and understand. So, I suggest making several smaller sentences so that it is easy to understand. 

Author Response

(The authors gave the same response as above.)
